# Quad Gaussian Networks for Vehicle Detection in Aerial Images

**DOI:** 10.3390/s24175661

**Published:** 2024-08-30

**Authors:** Haixiang Liang, Yuqing Wang

**Affiliations:** 1Changchun Institute of Optics, Fine Mechanics and Physics, Chinese Academy of Sciences, Dongnanhu Road 3888, Changchun 130033, China; lianghaixiang22@mails.ucas.ac.cn; 2University of Chinese Academy of Sciences, No. 1 Yanqihu East Rd, Huairou District, Beijing 101408, China

**Keywords:** object detection, vehicle detection, remote sensing, deep learning

## Abstract

Vehicle detection in remote sensing images is a crucial aspect of intelligent transportation systems. It plays an essential role in road planning, congestion control, and road construction in cities. However, detecting vehicles in remote sensing images is challenging due to their small size, high density, and noise. Most current detectors that perform well in conventional scenes fail to achieve better results in this context. Thus, we propose a quad-layer decoupled network to improve the algorithm’s performance in detecting vehicles in remote sensing scenes. This is achieved by introducing modules such as a Group Focus downsampling structure, a quad-layer decoupled detector, and the GTAA label assignment method. Experiments demonstrate that the designed algorithm achieves a mean average precision (mAP) of 49.4 and operates at a speed of 3.0 ms on the RTX3090 within a multi-class vehicle detection dataset constructed based on the xView dataset. It outperforms various real-time detectors in terms of detection accuracy and speed.

## 1. Introduction

Detecting and analyzing vehicle information from images forms the basis of intelligent transport [1]. With the growth of the global economy and the maturation of industrial systems in recent decades, the demand for vehicles has continued to increase, resulting in a significant rise in vehicle ownership [2]. This has inevitably led to numerous issues in urban transportation and has placed greater demands on intelligent transportation systems. Compared to traditional ground sensors, remote sensing cameras on UAVs or satellites have a wider field of view and can provide more target information, reducing costs and increasing efficiency [3]. This makes them more suitable for analyzing traffic scenarios. As a result, vehicle detection based on aerial images has gained attention and has been extensively studied in scenarios such as traffic monitoring, urban planning, and driver assistance [4].

Vehicle detection in aerial images presents several challenges. Firstly, remote sensing cameras have a wide shooting range, which can reach hundreds of meters to several kilometers [5]. This results in smaller vehicles within the image, making it difficult for algorithms to accurately extract target feature information and detect targets [6]. Secondly, remote sensing images have a more complex background space, with numerous confusing targets such as oil drums or containers [7]. This complexity makes it challenging for algorithms to differentiate between the target and the vehicles [8]. Additionally, in dense scenes like congested roadways or parking lots, accurately classifying each target to reduce errors or missed detections poses a difficulty in network design [9]. Finally, ensuring algorithm accuracy while maintaining real-time performance for practical applications is a challenging task.

In recent years, several methods have been proposed to address the above problems, and these methods can be divided into two groups: an improvement in network structures or an improvement in training methods. The former, such as FPNs [10,11,12], transformers [13,14], or super-resolution-based methods [15,16], generally improve the performance of algorithms significantly, but the disadvantage is that they usually increase the computational complexity of the network. The latter improves the training method by adjusting the loss or label assignment process to improve the final performance [17,18], and the advantage of these methods over the former is that they tend to not affect the running efficiency of the algorithm.

These methods can greatly improve the performance of vehicle detection. However, they are not able to overcome all the difficulties of vehicle detection. FPNs can improve the performance of the algorithm in small and dense target detection without affecting conventional object detection, but when encountering scenarios such as remote sensing vehicle detection, where the vast majority of targets are very small (less than 16 pixels), it may be difficult for the detection head to pick up the features of these tiny targets due to the high downsampling multiplicity in the existing FPN structure. Transformer or super-resolution-based approaches incur significant additional computational costs, making the algorithms difficult to deploy on low-power platforms. Many of the improvements to training methods, such as ATSS [17], can dramatically increase the time and memory cost of training the network.

To solve the above problems and ensure the real-time performance of the algorithm, we design a quad Gaussian network for vehicle detection in aerial images based on the C3 model of YOLOv5s. As shown in Figure 1, the network consists of three parts, where the backbone part consists of a C3 layer with a Group Focus downsampling layer, and the Quad Neck layer is based on the idea of FPN-PAN, which connects the different input layers from the backbone and outputs the fused features to their respective decoupled detectors. In addition, a label assigner and a loss function based on Gaussian Kullback–Leibler divergence are used to train the model. The main contributions of this paper are as follows:We propose a Group Focus method in order to improve the downsampling module in the backbone, which is able to take care of all the points in the feature map compared to a traditional strided convolution, and is able to reduce a loss of information from the small targets in the feature extraction process, which will help in the detection of vehicles with smaller scales in aerial images.Aiming at small and dense targets in vehicle detection, we design a Quad Decoupled Detection Head, which consists of a quad neck and decoupled head. The quad neck introduces a 4x downsampling structure in the FPN, which can allow more small target features to be input into the detector head. The decoupling of the target category and position prediction in the detector head can reduce mutual interference and, thus, increase the efficiency of feature utilization, which is helpful for detecting dense small targets.We design a Gaussian Task-Aligned Assigner for sample allocation by Kullback–Leibler divergence between prediction frames and actual frames, which assigns more appropriate prediction frames to dense small targets and reduces the cases where targets are ignored or have difficulty converging, thus improving target detection accuracy, especially for long-tail data.
Figure 1Overview of proposed methodology in our paper.
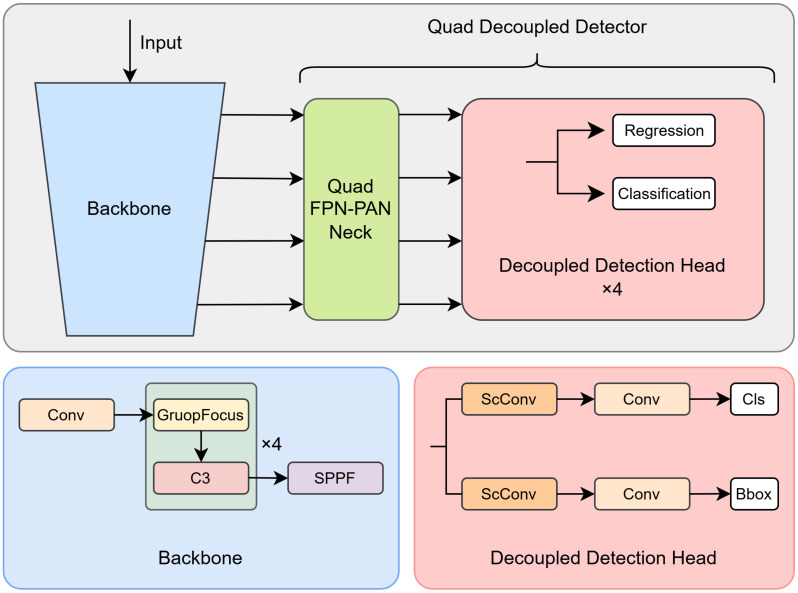


## 2. Related Works

### 2.1. Generalized Object Detectors

Generalized target detectors have made great progress after nearly a decade of development, and currently, there are mainly two-stage detection methods based on R-CNN [19] and one-stage detection methods based on YOLO [20] or RetinaNet [21]. Among them, the RCNN method was proposed in 2014, which adopts the candidate region generation method to obtain the target frame to be selected and uses a convolutional neural network and SVM for classification and target frame refinement. The subsequent Fast-RCNN [22] and Faster-RCNN [23] improved this method in terms of operational efficiency, including replacing SVM with CNN for target classification and proposing an RPN architecture to replace selective search.

These improvements substantially enhance the performance of RCNN networks. However, the real-time performance of RCNN is constrained by the two-stage detection architecture, so the one-stage detector is proposed. The one-stage detection network removes the RPN structure and directly outputs the detection results within the network by introducing anchors. So, the detection speed is faster and more suitable for real-time deployment in low-power scenarios. The most recent developments in the field of one-stage detectors are RTMDet [24], YOLOv9 [25], and YOLOv10 [26].

In recent years, detectors based on the attention mechanism have also been proposed. DETR [27] is based on the encoder–decoder structure of the transformer and label assignment by the Hungarian algorithm, which converts the target detection problem into a sequential output problem so as to achieve the end-to-end output of the detection results without the need for post-processing. Based on this idea, Zhao et al. [28] designed an RT-DETR enabling the DETR algorithm to run in real-time.

### 2.2. Small Object Detection

The difficulty of small target detection lies in how to effectively extract the feature information with less redundancy in the target. Feature Pyramid Networks [10] handle targets of different scales by receiving feature information from different downsampling layers in the backbone, and the combination of this structure with feature fusion modules, such as PAN, can simultaneously improve the accuracy of target detection algorithms on targets of different sizes.

Methods based on image enhancement were applied to small target detection. Bai et al. [15] proposed Mtgan to enhance small target detection by generating high-resolution targets, and Ma et al. [16] proposed to enhance infrared small target detection by generating labels and feature mapping.

In addition, attention-based methods have also been applied to small target detection. SCRDet [9] highlights small target regions in a feature map by designing a pixel attention layer and a channel attention layer, while attenuating the interference of noise. FBR-Net [29] designs a hierarchical attention-based mechanism that balances the features of different layers in a pyramid, thus enhancing the learning ability of small targets in complex situations.

Improving the training method of algorithms can also enhance the detection performance of small targets, such as the ATSS [17] method, which proposes an adaptive sample allocation strategy to batch better prediction frames for small target frames and, thus, enhance the performance of small target detection. Wang et al. [18] proposed a Gaussian Wasserstein distance instead of IoU for tiny target detection, which can be applied in sample allocation and loss function, which makes the algorithm tend to detect small targets, thus improving the algorithm’s detection performance of tiny targets, but will reduce the network’s ability to detect large targets [30].

### 2.3. Dense Object Detection

The goal of dense target detection is to distinguish between different targets and backgrounds in scenes where target frames are densely present and may overlap with each other. RetinaNet [21] is the most popular dense target detection method, which solves the problem of imbalance between foreground and background classes in dense scenes by reconstructing the standard cross-entropy loss, which allows the network to be more attentive to hard-to-detect targets, thus improving the algorithm’s ability to detect targets in dense environments.

Han et al. [31] proposed a Border-Align operator that utilizes boundary information between a target and a background in order to perform stronger classification and more accurate localization. Guo et al. [32] proposed a convex packet representation, which is based on deformable convolution to locate the extent of the object more accurately while mitigating feature aliasing to some extent. In addition, the two attention layers in the SCRDet method described above can similarly improve an algorithm’s detection ability in dense scenes.

### 2.4. Methods for Remote Sensing Vehicle Detection

In addition to the above generalized target detectors and improved schemes, some vehicle detection algorithms dedicated to aerial images have also been proposed. Zhong et al. [33] proposed a cascade detection method by combining two independent CNN models, where the first network is responsible for generating vehicle-like regions from feature maps at different scales, after which the generated regions are fed into the second network for feature extraction and decision-making operations.

Yu et al. [34] proposed a convolutional capsule network, where the algorithm first segments the input image into multiple hyper pixels to generate meaningful and meaningless patches, which are subsequently fed into the capsule network and labeled as vehicles or background. Ji et al. [35] were influenced by super-resolution convolutional neural networks to train both SRCNN and vehicle detection end-to-end by generating an adversarial network and back-propagating the training loss into the SRCNN. The method is capable of generating high-resolution vehicle features in an unsupervised manner, thereby improving accuracy.

In order to effectively utilize useful information in multi-source data for better vehicle detection, Wu et al. [36] proposed a multi-source active fine-tuning vehicle detection (Ms-AFt) framework that can unify migration learning, target detection, and instance segmentation, which can exhibit better detection accuracy and generalization ability.

Kong et al. [37] proposed a two-stage detection framework with a parallel RPN structure designed to mitigate the scale variation problem by means of a density-based sample allocator in order to reduce low-quality sample allocation. In addition, a scale-based NMS was proposed to filter redundant suggestions hierarchically from different levels of the feature pyramid, and the algorithm was able to achieve better detection performance on multiple datasets.

For aerial vehicle detection in infrared scenes, Zhang et al. [38] proposed a DTNet that uses a learnable low-pass filter to decompose feature maps into low-frequency and high-frequency components. The interference of the optical system and background thermal radiation on the high-frequency components of the feature maps is eliminated, while the details and textures of the vehicles are preserved. Zhao et al. [14] proposed a YOLO-ViT-based method to solve the vehicle detection problem in infrared images by combining vit with the yolov7 framework in order to reduce the amount of floating-point computation and the parameters of YOLO and improve the accuracy to a certain extent.

## 3. Proposed Method

This chapter will provide a comprehensive account of the methodology depicted in Figure 1, which encompasses the Group Focus downsampling method, the quad-layer decoupled detection head, and the Gaussian Task-Aligned Assigner.

### 3.1. Group Focus Downsampling

Object detection algorithms typically use stepwise convolution or maximum pooling operations to downsample and output feature maps at different scales for detecting objects at different scales, which can result in the loss of fine-grained features in an image and, thus, reduce the detection accuracy of the algorithm [39]. In most detection tasks with good image resolution and moderate target sizes, this design problem does not arise. This is because the information in these scenarios is usually more redundant, and a certain degree of fine-grained loss will not lead to a degradation in object detection performance. However, detecting these small objects, which inherently have less feature information and lower signal-to-noise ratios, can be challenging due to the lack of extraction of fine-grained information. This can affect the algorithm’s ability to learn the overall target features and, consequently, the accuracy of the target detection algorithm.

The FOCUS structure is considered to be a method that can address the loss of information during downsampling. As shown in Figure 2, this module is based on the idea of image compression, which divides the input feature map into four parts, combines them vertically, and then uses group convolution to perform a convolution operation on each segmented small feature map separately to obtain the final downsampled feature map. Compared with strided convolution, this structure is able to sample all points of the input feature map, which avoids a loss of information in the downsampling process to some extent and, at the same time, attenuates the overall effect of noise on the image.

However, the FOCUS structure has the disadvantage of larger computational volume; compared with the traditional downsampling algorithm, FOCUS will bring 4 times the computational volume, which will affect the detection speed of the algorithm. In order to solve this problem, we designed the Group Focus structure based on the idea of group convolution, as shown in Figure 3, which is divided into a separated focus layer and an adaptive connection layer. The separated focus layer performs the convolution operation on the compressed subgraphs separately and combines them for downsampling; this separated operation reduces the computational amount, but it may cause the model to lose the extraction of the overall features of the image, so we introduce an adaptive connection layer to add the information of the original image for the downsampling process.

### 3.2. Quad-Layer Decoupled Detection Head

Most target detection algorithms use an FPN structure in the neck layer to input and fuse 8/16/32 times compressed feature maps. This design allows the algorithm to detect targets of different sizes, enhancing overall performance. However, it is important to note that most vehicle targets in remote sensing imagery are only 10–16 pixels in size, so excessive sampling can result in a loss or significant degradation of feature information in small targets, which, in turn, can lead to a reduction in the feature information obtained by the algorithm in the neck layer, ultimately affecting feature fusion.

Therefore, we designed a quad-layer FPN structure that adds an additional 4x downsampling feature map input for detecting tiny vehicle targets without compromising the algorithm’s ability to detect objects at other scales. We also introduce a decoupling header, which computes the class and position of the target separately through different networks, thus avoiding interference between different tasks.

To further improve the detection ability of the detector head for small targets in dense scenes, we propose a decoupled detector head (DDH), as shown in Figure 1, where we decouple the category prediction and box prediction of the target, and this decoupled structure can prevent the prediction of the target’s category and box from interfering with each other, thus increasing the accuracy of classification and regression at the same time.

In addition, we introduce a spatial and channel reconstruction convolution [40] (ScConv) module to further improve detection accuracy and reduce computational effort. As illustrated in Figure 4, ScConv consists of two units—spatial reconstruction and channel reconstruction—which can reduce computational complexity and improve performance by taking advantage of spatial redundancy and channel redundancy in the algorithm and reconstructing input feature maps, thus allowing us to provide fewer parametric feature maps with better expressive ability.

### 3.3. Gaussian Task-Aligned Assigner

We believe that the major cause of incorrect sample allocation is the flaws in IoU-based target frame representation for small target detection. There are two problems with IoU in tiny object detection.

Firstly, as shown in Figure 5, for objects at different scales, the decreasing trend of IoU for tiny targets will be much higher than that for large objects under the same degree of deviation in the prediction frame from the ground truth, which will negatively affect the algorithm’s allocation of samples and the calculation of the loss function [30].

Secondly, the IoU is unable to represent the distance information between two non-overlapping boxes, which may lead to these objects being assigned to sub-optimal target boxes or not being assigned a target box during a sample allocation session for tiny targets. As shown in Figure 6, in this scenario, gt1 is assigned to a sub-optimal larger target box, while gt2 cannot be accurately assigned to a suitable positive sample because there is no overlapping detection box with it, resulting in the algorithm being difficult to train on this target in the subsequent process.

To tackle these issues, we design a Gaussian Task-Aligned Assigner and the corresponding loss function. We first model all the bounding boxes B(x, y, w, h) as a two-dimensional Gaussian distribution modeled as N(μ, Σ) with the following transformation relation:(1)$$\mu =\left[\begin{array}{c} c_{x}\\ c_{y} \end{array}\right],\Sigma =\left[\begin{array}{cc} \frac{w^{2}}{4} & 0\\ 0 & \frac{h^{2}}{4} \end{array}\right].$$

There are three ways to measure the similarity between two Gaussian distributions: Wasserstein distance (GWD), Kullback–Leibler dispersion (KLD), and Bhattacharyya distance (BCD). We will describe these three different approaches and analyze their advantages and drawbacks.

#### 3.3.1. Wasserstein Distance

The Wasserstein distance is derived from the theory of optimal transport. For two Gaussian distributions $N_{t}$($\mu_{t}$, $\Sigma_{t}$) and $N_{p}$($\mu_{p}$, $\Sigma_{p}$), the Wasserstein distance between them is as follows:(2)$$\begin{array}{cc} GWD(N_{p},N_{t})= & ∥\mu_{p}-\mu_{t}{∥}_{2}^{2}+{∥\Sigma_{p}^{1/2}-\Sigma_{t}^{1/2}∥}_{F}^{2}\\ = & {\left(x_{p}-x_{t}\right)}^{2}+{\left(y_{p}-y_{t}\right)}^{2}+\frac{{\left(w_{p}-w_{t}\right)}^{2}+{\left(h_{p}-h_{t}\right)}^{2}}{4} \end{array}$$

The Wasserstein distance is characterized by its sensitivity to smaller target frame errors, allowing it to predict and train more accurately on small targets compared to IoU, and when there is no overlap between the target frames, the GWD is still able to accurately represent the similarity between the two. However, because the formula contains the Euclidean distance of the target centroid, the GWD is not scale-invariant, so when used as a loss function, the distance needs to be normalized, and one way of doing this is as follows:(3)$$GWD_{Norm}=1-\frac{1}{2+\sqrt{GWD}}$$

Although the normalized GWD can alleviate its scale invariance problem, the overall loss is still not scale-invariant. In addition, the assigned object frames may not be optimal for targets with large scales in the label assignment session, leading to the fact that the method may reduce the algorithm’s ability to detect non-tiny objects [41].

#### 3.3.2. Kullback–Leibler Dispersion and Bhattacharyya Distance

In addition to GWD, the other two metrics are Kullback–Leibler dispersion (KLD) and Bhattacharyya distance (BCD). The formulae for KLD between two Gaussian distributions are as follows:(4)$$\begin{array}{cc} KLD(N_{p},N_{t})= & \frac{1}{2}{\left(\mu_{p}-\mu_{t}\right)}^{⊤}\Sigma_{t}^{-1}\left(\mu_{p}-\mu_{t}\right)+\frac{1}{2}\mathrm{Tr}\left(\Sigma_{t}^{-1}\Sigma_{p}\right)+\frac{1}{2}\ln\frac{|\Sigma_{t}|}{|\Sigma_{p}|}-1\\ = & \frac{1}{2}\left(\frac{w_{p}^{2}}{w_{t}^{2}}+\frac{h_{p}^{2}}{h_{t}^{2}}+\frac{4\Delta^{2}x}{w_{t}^{2}}+\frac{4\Delta^{2}y}{h_{t}^{2}}+\ln\frac{w_{t}^{2}}{w_{p}^{2}}+\ln\frac{h_{t}^{2}}{h_{p}^{2}}-2\right) \end{array}$$

The formula for BCD is as follows:(5)$$BCD\left(N_{p},N_{t}\right)=\frac{1}{8}{\left(\mu_{p}-\mu_{t}\right)}^{⊤}\Sigma^{-1}\left(\mu_{p}-\mu_{t}\right)+\frac{1}{2}\ln\frac{\det(\Sigma )}{\sqrt{\det(\Sigma_{p}\Sigma_{t})}}$$

The advantage of KLD over GWD is that it exhibits better scale invariance, rendering it a more suitable choice as a loss-time training function. The disadvantage of KLD is that for two target frames that do not overlap, the similarity represented by KLD is inferior. This is in contrast to GWD. However, due to the advantage of scale invariance, KLD is preferable in the label assignment session [42]. BCD is similar to KLD for the majority of scenarios. Similarly, the distinction lies in the fact that BCD has symmetry.

#### 3.3.3. Design of Gaussian Task-Aligned Assigner

In summary, we design a Gaussian Task-Aligned Assigner based on the KLD and TAA [43]. This assigner selects positive samples based on the weighted value of the scores of classification and regression. For each real box, K positive samples are selected to regulate the loss of the target box and category according to the weighted scores. The weighted scores are computed as shown below:(6)$$score=s^{\partial }\times d^{\beta },$$
where s is the prediction category score, d is the KLD score of the prediction box and ground truth, and α and β are the weights of the hyper parameters.

## 4. Experimental Results

### 4.1. Datasets and Implementation Details

To verify the effectiveness of the proposed method in this paper, we constructed a remote sensing vehicle detection dataset based on the xView datasets. The xView [44] dataset is one of the largest aerial remote sensing object detection datasets available. It includes over 800 remote sensing images from around the world, captured using a WorldView3 camera with a resolution of 0.3 m. The total area covered by the dataset is more than 1400 square kilometers, and it contains over 1 million labeled instances of 60 classes of targets.

Since the original images have high resolution, we use the sliding window method to segment the images into subgraphs of 1000 × 1000 resolution with an overlap rate of 0.2. The total number of segmented images is 10,000. All the vehicle targets in the images are filtered and classified into the three categories of cars, buses, and trucks, and the total number of filtered targets is 0.87 million. The size of the car class ranges from 8–16 pixels with an average of 12 pixels, while the size of the bus and truck ranges from 15–30 pixels with an average of 21 pixels. The details of the dataset are shown in Table 1.

All our experiments were conducted on Ubuntu with a computer containing an i7-10700k CPU model, 32G DDR4 RAM, an NVIDIA GEFORCE RTX 3090 GPU model (Nvidia, Santa Clara, CA, USA), Python 3.9 programming language, the PyTorch 1.13.1 deep learning framework, and CUDA version 11.2. In terms of training parameters, we used an SGD optimizer with the momentum set to 0.937, an initial learning rate of 0.01, a weight decay of 0.0005, a batch size of 4, and a training batch of 200 rounds, and the parameters of GTAA are as follows: α is 1, β is 6, and k is 9. All of our experiments were not loaded with pre-training weights.

Figure 7 shows the loss during the training of the algorithm versus the mAP curve within the val dataset, with the dfl loss eventually converging to 1.053 and reaching a mAP of 0.48 on the validation set.

### 4.2. Evaluation Metrics

We used mean average precision (mAP) and average processing time per image (ms) as indicators. The formulae for calculating P, R, and mAP are as follows:(7)$$P=\frac{TP}{TP+FP}$$
(8)$$R=\frac{TP}{TP+FN}$$
(9)$$AP=\int_{0}^{1}P\left(R\right)dR$$
(10)$$\mathrm{mAP}=\frac{\sum_{i=1}^{k}{\mathrm{AP}}_{i}}{k}$$
where TP is the number of accurate detections, FP is the number of false detections, FN is the number of missed detections, P is the precision rate, R is the regression rate, and k represents all detection categories. mAP’s thresholds use two metrics—AP50 and AP50-95—with the former using 50% IoU as the accurate detection threshold to compare the algorithm’s performance in detection, while the latter is based on the fact that at 50–95% IoU, it is able to verify accuracy at higher target frame overlap rates.

### 4.3. Ablation Experiments

In order to ascertain the efficacy of each module proposed in this paper, we conduct ablation experiments on each module proposed in Chapter 3; we use yolov5s as the baseline for the ablation experiments, the results of which are presented in Table 2.

As shown in Table 2, the first group is the baseline, with a mAP50 value of 38.5%, a mAP50-95 value of 14.8%, and a computation speed of 1.7 ms on RTX3090. In the second group, after adding the Group Focus downsampling structure to the backbone, the mAP50 improves by 2.7% and mAP50-95 improves by 0.9%, which improves the detection performance of the three categories, while the algorithm shows a slight increase in computation time, so it can be concluded that our proposed downsampling module can improve the feature extraction ability of the backbone.

The third group is based on group 2 and introduces GTAA, which improves the mAP50 by 0.8% and the mAP50-95 by 0.3%, where the bus and truck classes are improved by 0.9% and 1.3%, respectively, and the algorithms are not reduced in size or operational efficiency because improving the training method and the loss function does not increase the inference cost of the network. The fourth group improves to 48.2% for mAP50 and 20.3% for mAP50-95 based on the introduction of the Quad Decoupled Detection Head in the second group. Furthermore, the time required for the algorithmic computation increases. The fifth group combines all the modules proposed in this paper, and the final mAP50 is 49.4% and the final mAP50-95 is 20.6%, with a computation time of 3 ms.

The fifth and third groups demonstrate that GTAA has limited improvement on the car class but is able to improve the performance of the bus and truck classes with smaller sample sizes, thus improving the accuracy of the algorithm. We speculate that GTAA can allow these small targets with small sample sizes to participate more in training, avoiding convergence difficulties due to irrational sample allocation.

It is imperative to demonstrate that the Quad Decoupled Detection Head is an effective standalone module rather than merely showcasing its enhanced potential through parameter optimization, which would be a circular argument. To this end, the backbone of the baseline has been expanded in order to attain a comparable parameter count to that of the current method. The resulting performance is presented in the sixth group, exhibiting a mAP50 of 45.5% and a mAP50-95 of 18.9%, with a computation time of 3.4 ms. This represents a significant improvement on the previous group, providing further evidence in support of the proposed approach.

In view of the evidence that the detection head has the potential to deliver considerable improvements in performance, additional trials were conducted on the three sub-modules of the quad neck, decoupled head, and ScConv, with the objective of providing a more comprehensive validation of the proposed methodology. The experiments are based on the second group in Table 2 as the baseline, with all other experimental parameters remaining unchanged. The results of the experiment are presented in Table 3.

As illustrated in Table 3, the incorporation of quad-layer and decoupled heads was observed to increase the mAP50 to 45.6% and 44.9%, respectively. It is notable that the quad layer has demonstrated a more pronounced impact on the car class, whereas the decoupled head exhibited a more substantial improvement in the performance of the bus and truck classes.

Following the combination of the two methods, the performance of the car class exhibits a discernible decline in comparison to the quad layer, and the overall mAP of the algorithm nevertheless demonstrates an improvement, reaching 47%. It can, thus, be assumed that the quad layer enhances the algorithm’s overall capacity to detect small targets, whereas the Decoupled Head separates target position prediction from category prediction, thereby favoring long-tailed targets. However, this approach may potentially have an adverse impact on other categories. Finally, the introduction of ScConv increases the AP by 0.8% and reduces the number of parameters and computation time, indicating that ScConv is capable of enhancing performance while reducing computation.

### 4.4. Comparison Results

To evaluate the effectiveness of our method, we compared it with YOLOV6s [45], YOLOv8s, YOLOv8n, Swin-Transformer-Tiny [46], RTDETR [28], and LSKNet [47] in terms of mAP and computational speed using the xView vehicle detection dataset. Swin-Transformer-Tiny uses the same neck and head as YOLOv8s, and RT-DETR uses the same backbone as our method.

Table 4 shows that our proposed method achieves a mAP of 49.4, with a computational speed of 3.0 ms, which is significantly higher than the Swin-Transformer-Tiny, RTDETR-YOLO, and LSKNet methods in terms of both accuracy and speed. The LSKNet method has the lowest accuracy, while the Swin-Transformer-Tiny method has the longest running time and does not increase mAP. The YOLOv6s algorithm is comparable to our method in terms of computational speed but falls short in overall and per-category computational accuracy.

YOLOv8s is slightly better than the algorithm proposed in this paper in terms of accuracy and has an advantage in detecting both bus and truck categories. However, it takes longer to compute, being 47% slower than our method, which may affect the deployment of the algorithms on platforms with lower computational power. Moreover, in the car category with the smallest target size, YOLOv8s does not have the advantage of accuracy. When compared to YOLOv8n, which has the same network structure but a smaller backbone, our method shows significantly better accuracy metrics and outperforms it, despite being slightly slower.

### 4.5. Visualization of Results

Figure 8 shows a set of visualized comparison experimental results, we can see that YOLOv8n, YOLOv6s, Swin-Tiny, RT-DETR, and LSKNet are sub-optimal in terms of detection accuracy in the car parks and roads where the objects are densely packed in the lower left of the image, where YOLOv8n and YOLOv6s perform the worst and capture fewer vehicles in car parks. Our method is close to YOLOv8s in terms of accuracy.

In the intersection below the car park, YOLOv6s, YOLOv8n, and LSKNet can only detect fewer vehicles, and YOLOv8s, RT-DETR, and our method are able to detect the vast majority of objects but not as well as Swin-Tiny, in terms of regression rate. On the houses on the right side of the intersection, the algorithms in this paper show more false detections than YOLOv8s and RT-DETR.

In the two car parks in the middle of the image, the regressions of YOLOv6s, YOLOv8n, and LSKNet are poor, with YOLOv6s also showing more aliasing. The remaining algorithms have close detection accuracies, but the method in this paper and YOLOv8s both show some aliased frames, RT-DETR has a slightly higher number of missed detections, and swin-t obtains the best performance.

For the road scene in the lower left corner, this paper’s algorithm achieves the best detection performance, and all methods except YOLOv8s show different degrees of leakage, while this paper’s method is better than YOLOv8s in terms of frame prediction accuracy.

It is known that lighting conditions greatly influence the efficiency of vehicle detection. Therefore, to visually compare the detection performance of different algorithms in low-light conditions, we chose an image with dark-light conditions. As shown in Figure 9, although Swin-Tiny and RT-DETR are able to achieve good detection accuracy under normal conditions, these methods’ abilities to detect targets in low-light scenarios show a sharp decline, with a large number of missed detections, seriously affecting the algorithms’ performances on the entire dataset.RT-DETR detected large targets that were not monitored by all other algorithms.

LSKNet also shows a large drop in detection capability and is outperformed in accuracy by YOLOv6s vs. YOLOv8n. YOLOv8n and YOLOv6s have similar detection capabilities, but YOLOv8n has poorer target frame refinement, which affects the mAP at higher metrics.

Comparing our method with YOLOv8s, the detection performance of the two is close in the vast majority of scenes, but YOLOv8s detects more targets in the center of the left side. However, YOLOv8s showed more aliasing in detection, which had a negative impact on precision.

## 5. Conclusions and Discussion

To address the challenges of detecting small objects with high density and noise interference in remote sensing vehicles, we developed a quad-layer decoupled network using YOLOv5s. By designing the Group Focus downsampling module and introducing the quad-layer decoupled detection head and Gaussian Task-Aligned Assigner, we improved the performance of the algorithm for vehicle detection in remote sensing images. It was experimentally demonstrated that the designed algorithm achieved a mAP of 49.4 on the vehicle detection dataset constructed based on xView, with a running time of 3.0 ms on RTX3090, which exceeds a variety of real-time detectors in terms of accuracy and computational speed.

In future work, we will try to design a real-time vehicle detector based on the structure of DETR. DETR uses the Hungarian algorithm to perform sample assignments in a one-to-one form and converts the object detection problem into a sequential output problem, which can effectively enhance object detection in dense scenarios. We will focus on improving the DETR’s object detection performance and detection efficiency on small objects, thereby creating a real-time vehicle detector with both high accuracy and efficiency. 

## Figures and Tables

**Figure 2 sensors-24-05661-f002:**
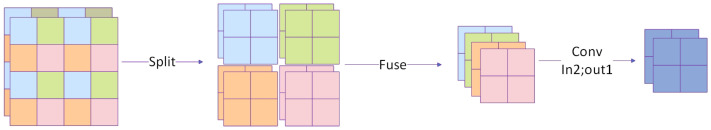
The focus downsampling structure.

**Figure 3 sensors-24-05661-f003:**
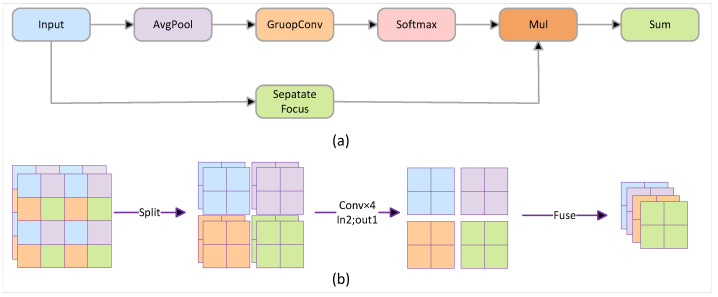
Proposed Group Focus downsampling structure. (**a**) Group Focus structure diagram divided into separated focus module and adaptive connection layer. (**b**) The separated focus module diagram.

**Figure 4 sensors-24-05661-f004:**
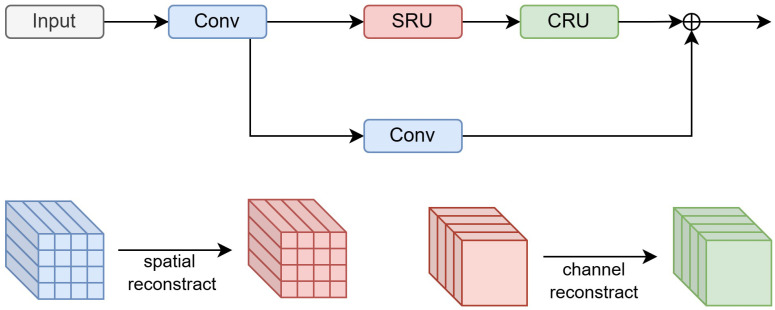
A structural diagram of ScConv. The input feature maps are spatially reconstructed and channel reconstructed, and residual concatenation is introduced.

**Figure 5 sensors-24-05661-f005:**
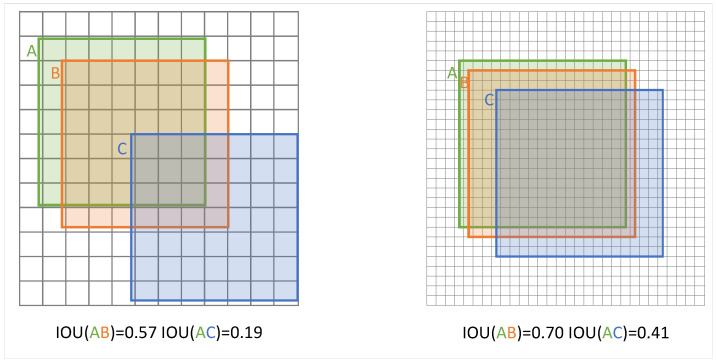
Sharp decreasing trend of IoU in tiny object detection.

**Figure 6 sensors-24-05661-f006:**
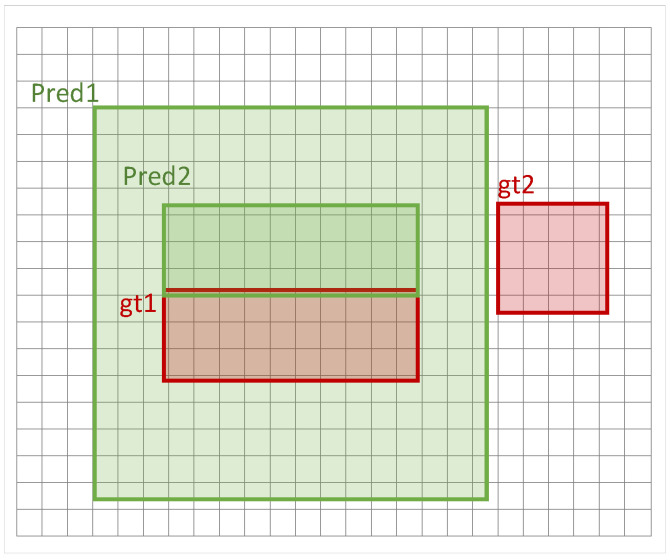
Disadvantages of IoU in tiny object labeling assignment.

**Figure 7 sensors-24-05661-f007:**
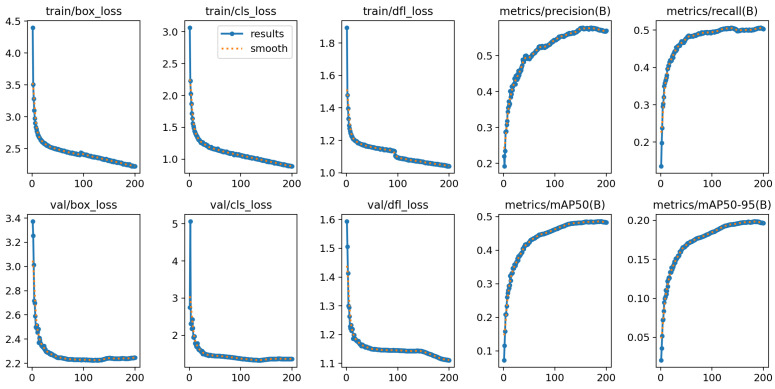
A graph of the training results.

**Figure 8 sensors-24-05661-f008:**
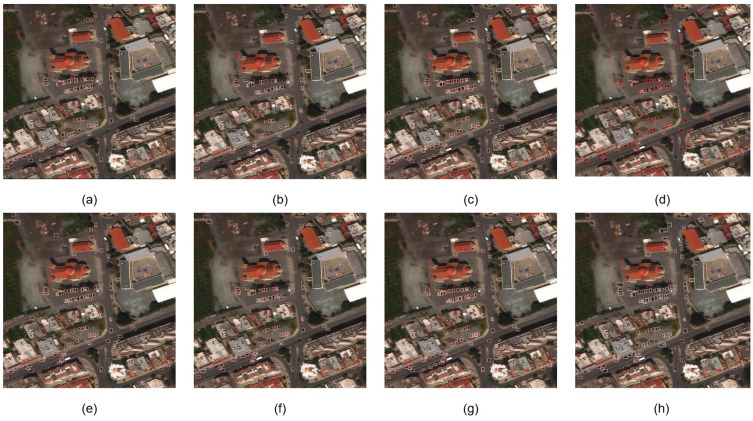
Visualizations of different network models. (**a**) YOLOv6s; (**b**) YOLOv8n; (**c**) YOLOv8s; (**d**) DETR-YOLO; (**e**) Swin-Tiny; (**f**) LSKNet; (**g**) ours; (**h**) ground truth.

**Figure 9 sensors-24-05661-f009:**
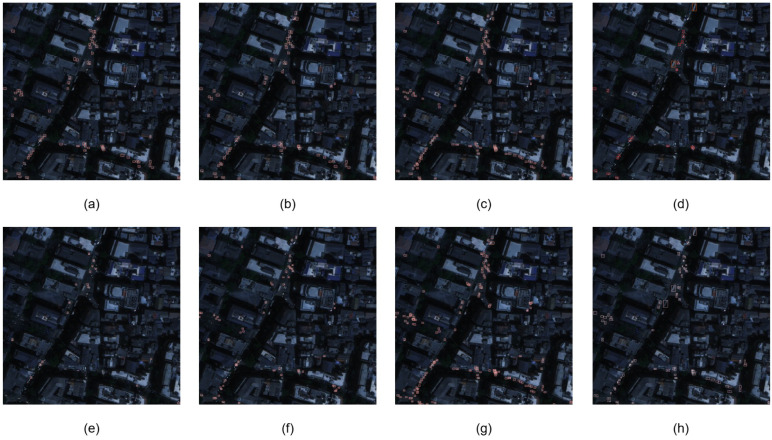
Visualization of results in low-light conditions. (**a**) YOLOv6s; (**b**) YOLOv8n; (**c**) YOLOv8s; (**d**) DETR-YOLO; (**e**) Swin-Tiny; (**f**) LSKNet; (**g**) ours; (**h**) ground truth.

**Table 1 sensors-24-05661-t001:** Overview of xView vehicle detection dataset.

	Images	Bus	Car	Truck	Total
Train	15,590	21,086	573,092	93,413	687,591
Val	1283	3491	98,037	15,163	116,691
Test	1836	4201	54,350	13,165	71,696
Total	18,709	28,778	725,479	121,741	875,998

**Table 2 sensors-24-05661-t002:** Results of ablation experiments.

Metrics	mAP50%	mAP50-95/%	APcar/%	APtruck/%	APbus/%	Speed/ms
Baseline	38.5	14.8	53.7	27.9	33.9	1.7
GFocus	41.2	15.7	56.1	30.1	37.5	1.9
GFocus + GTAA	42.0	16.0	56.2	31.4	38.4	1.9
GFocus + QDD	48.2	20.3	63.0	38.4	43.1	3.0
All	49.4	20.9	63.1	38.8	46.3	3.0
Extended Net	45.5	18.9	60.7	35.6	40.3	3.4

**Table 3 sensors-24-05661-t003:** Results of ablation experiments of Quad Decoupled Detection Head.

Metrics	mAP50%	mAP50-95/%	APcar/%	APtruck/%	APbus/%	Speed/ms
Baseline	41.2	15.7	56.1	30.1	37.5	1.9
Quad layer	45.3	17.1	63.7	35.3	36.9	2.6
Decoupled head	44.9	17.8	58.6	35.6	40.5	2.4
Combined heads	47.4	19.8	62.6	36.7	42.8	3.2
Heads + ScConv	48.2	20.3	63.0	38.4	43.1	3.0

**Table 4 sensors-24-05661-t004:** Results of comparison experiments.

Method	mAP50/%	mAP50-95/%	APcar/%	APtruck/%	APbus/%	Speed/ms
YOLOv6s	38.7	16.7	52.9	30.7	32.5	3.1
YOLOv8s-p2	49.8	20.4	66.0	39.0	44.5	4.4
YOLOv8n-p2	39.5	19.0	54.5	32.9	31.2	2.5
Swin-Tiny	40.2	17.5	54.7	32.6	33.4	11.7
LSKNet	33.5	14.2	45.1	29.2	26.2	6.0
RT-DETR	42.8	18.0	65.3	32.3	31.0	5.5
Ours	49.4	20.9	63.1	38.8	46.3	3.0

## Data Availability

The xView dataset used in this article is an open-source dataset and is available at http://xviewdataset.org/.

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
