# Peer review of "Quad Gaussian Networks for Vehicle Detection in Aerial Images"

_sensors, 2024, doi:10.3390/s24175661_

Round 1

Reviewer 1 Report

Comments and Suggestions for Authors

In this paper, the authors propose a novel network for the small object (i.e., vehicle) detection in aerial images. Several concerns need to be addressed during this review round.

1. Simplification of Methodology Diagram

The flowchart detailing the methodology contains an excessive level of complexity. It would be beneficial to streamline the diagram by omitting elements that are considered common knowledge within the field, thus allowing for a clearer presentation of the unique aspects of the proposed method.

2. Inclusion of Ablation Study

The manuscript would greatly benefit from the addition of an ablation study. This would serve to demonstrate the effectiveness of the individual components introduced in this work, such as the Group Focus downsampling structure, the quad-layer decoupled detection head, and the Gaussian TAA label assignment method.

3. Enhancement of Figures 7 and 8

To better highlight the advantages of the proposed method, it is recommended to include ground truth data in Figures 7 and 8 for comparison. The current visual representation does not effectively convey the superiority of the method.

4. Update of References

The references cited in the manuscript appear to be outdated. It is advisable to incorporate more recent advancements in deep learning within the related work section.

[1] Deep learning for unmanned aerial vehicles detection: A review.

[2]SWCARE: Switchable learning and connectivity-aware refinement method for multi-city and diverse-scenario road mapping using remote sensing images

Author Response

Comments 1:[Simplification of Methodology Diagram: The flowchart detailing the methodology contains an excessive level of complexity. It would be beneficial to streamline the diagram by omitting elements that are considered common knowledge within the field, thus allowing for a clearer presentation of the unique aspects of the proposed method.]

Response 1:[Agree. We have modified the flowcharts in Figures 1 to 4 by simplifying the elements in the diagrams to highlight the improvements. In addition, we redesigned the network to incorporate the ScConv module into Neck as well, and the map50 of the algorithm went up slightly by 0. 5% and the computation time went down to 3.0ms.]

Comments 2:[Inclusion of Ablation Study: The manuscript would greatly benefit from the addition of an ablation study. This would serve to demonstrate the effectiveness of the individual components introduced in this work, such as the Group Focus downsampling structure, the quad-layer decoupled detection head, and the Gaussian TAA label assignment method.]

Response 2:[Thank you for pointing this out. Therefore, We have supplemented the ablation experiments section in Section 4.3 of the article to verify the respective effectiveness of each module and to analyse the causes. We also performed additional ablation experiments on all sub-modules of the Quad Decoupled Detection Head module to further validate the effectiveness of the module.]

Comments 3:[Enhancement of Figures 7 and 8: To better highlight the advantages of the proposed method, it is recommended to include ground truth data in Figures 7 and 8 for comparison. The current visual representation does not effectively convey the superiority of the method.]

Response 3:[Agree. We have included ground truth as a comparison in Figures 7 and 8 to be used to demonstrate the superiority of the method.]

Comments 4:[Update of References: The references cited in the manuscript appear to be outdated. It is advisable to incorporate more recent advancements in deep learning within the related work section.

[1] Deep learning for unmanned aerial vehicles detection: A review.

[2]SWCARE: Switchable learning and connectivity-aware refinement method for multi-city and diverse-scenario road mapping using remote sensing images.]

Response 4:[The citations presented are articles related to remote sensing vehicle detection and intelligent transport, which we believe would be beneficial to cite for the work in this paper, so we cite them in Chapter 1. We also additionally cite the following articles to demonstrate recent advances in deep learning within the related work section.

[1]Wang C Y, Yeh I H, Liao H Y M. Yolov9: Learning what you want to learn using programmable gradient information[J]. arXiv preprint arXiv:2402.13616, 2024.

[2]Wang A, Chen H, Liu L, et al. Yolov10: Real-time end-to-end object detection[J]. arXiv preprint arXiv:2405.14458, 2024.

[3]Zhang N, Liu Y, Liu H, et al. DTNet: A Specialized Dual-Tuning Network for Infrared Vehicle Detection in Aerial Images[J]. IEEE Transactions on Geoscience and Remote Sensing, 2024.

]

Reviewer 2 Report

Comments and Suggestions for Authors

1. The paper needs to further clarify why existing networks are insufficient for small object detection in Aerial Images and theoretically elaborate on the advantages of the improved network structure.

2. It is recommended to conduct ablation experiments to verify the specific contributions of each component and their impact on actual performance, thus confirming the effectiveness of these modules.

3. The author needs to explain the training, validation, and test set division ratios of the XView data. It is recommended to display the loss and mAP curves of the training process.

4. In the model performance comparison section, it is necessary to clarify whether the compared models used pre-trained weights to reasonably demonstrate the performance improvement of the modified model

Comments on the Quality of English Language

The paper is well-organized with a complete structure. The English expression is good, effectively presenting the entire research process.

Author Response

Comments 1:[The paper needs to further clarify why existing networks are insufficient for small object detection in Aerial Images and theoretically elaborate on the advantages of the improved network structure.]

Response 1:[Agree. We further elaborate the shortcomings of the current major algorithms in the direction of vehicle detection in aerial images in Chapter 1 and describe the motivation and advantages of the network structure designed in this paper. Modifications are highlighted. Furthermore, we redesigned the network to incorporate the ScConv module into Neck as well, and the map50 of the algorithm went up slightly by 0. 5% and the computation time went down to 3.0ms.]

Comments 2:[It is recommended to conduct ablation experiments to verify the specific contributions of each component and their impact on actual performance, thus confirming the effectiveness of these modules.]

Response 2:[Thank you for pointing this out. Therefore, We have supplemented the ablation experiments section in Section 4.3 of the article to verify the respective effectiveness of each module and to analyse the causes. We also performed additional ablation experiments on all sub-modules of the Quad Decoupled Detection Head module to further validate the effectiveness of the module.]

Comments 3:[The author needs to explain the training, validation, and test set division ratios of the XView data. It is recommended to display the loss and mAP curves of the training process.]

Response 3:[Thank you for pointing this out. We segmented the training/validation/testing set of 840 original images with labels in xView at a ratio of 8:1:1. Since the original images were too large, these images were segmented into sub-images of 1000*1000 pixels. Each dataset was segmented with a different number of subgraphs due to the different resolution of each original image. In addition, some of the sub-images did not contain vehicle targets, so these images needed to be cleaned. To prevent possible data leakage from subgraphs from the same parent image, we did not further integrate and segment the data. The final data ratio is shown in Table 1, the ratio of training set to other data sets is about 5:1. the ratio of test and validation set is about 3:2. Moreover, we have shown the loss function versus mAP variation curves during training in Fig. 7.]

Comments 4:[In the model performance comparison section, it is necessary to clarify whether the compared models used pre-trained weights to reasonably demonstrate the performance improvement of the modified model]

Response 4:[All of the ablation and comparison experiments we conducted were not loaded with pre-training weights and a description of this is added in Section 4.1. ]

Reviewer 3 Report

Comments and Suggestions for Authors

Comments to Manuscript sensors-3009615

In this paper, a quad-layer decoupled network is proposed to improve the algorithm’s performance in detecting vehicles in remote sensing scenes. By introducing modules such as the Group Focus downsampling structure, the quadlayer decoupled detector, and the GTAA label assignment method, the designed algorithm achieves good performance in terms of detection accuracy and speed for tiny object detection. Some comments are given below to improve the paper quality.

1. In my opinion, the Introduction Section of this manuscript is verbose and not focused. It should be modified with the related works added. Introducing the existing methods related to the subject is necessary, but the structure of the manuscript is not good. What is the author's purpose in separating the first two sections? At the same time, there must be more discussion about the innovation of the paper and the structure of the paper also needs to be introduced in the first section.

2. In page 5, the author introduces that the proposed method is based on YOLOv5s. However, we did not see any content about YOLOv5s through the article. It needs to be clarified whether the author rewrote the entire algorithm or only part of it. If it is the latter, the proportion needs to be clarified.

3. In Figure 6 and Figure 7 related contents, different IoUs may result in different experiment data. What is the basis for the author to choose these specific IoU values?

All in all, this is a nice piece work in a clear form and the research subject is cutting edge. A major revision is suggested by the reviewer.

Author Response

Comments 1:[In my opinion, the Introduction Section of this manuscript is verbose and not focused. It should be modified with the related works added. Introducing the existing methods related to the subject is necessary, but the structure of the manuscript is not good. What is the author's purpose in separating the first two sections? At the same time, there must be more discussion about the innovation of the paper and the structure of the paper also needs to be introduced in the first section.]

Response 1:[We have modified the Introduction section to focus on presenting the shortcomings of the current methodology as well as the motivations and innovations of the methods proposed in this paper. In addition, the introductory section on the structure of the paper has been similarly placed in the first chapter. The motivation for separating the first two chapters is that we noticed that most of the deep learning related articles published in this journal have adopted this lineage.]

Comments 2:[In page 5, the author introduces that the proposed method is based on YOLOv5s. However, we did not see any content about YOLOv5s through the article. It needs to be clarified whether the author rewrote the entire algorithm or only part of it. If it is the latter, the proportion needs to be clarified.]

Response 2:[Thank you for pointing this out. We designed the algorithm based on the backbone of YOLOv5s and modified the downsampling module in the backbone.YOLOv5s was also used in the baseline of the ablation experiments to demonstrate the effectiveness of the three modules designed.]

Comments 3:[In Figure 6 and Figure 7 related contents, different IoUs may result in different experiment data. What is the basis for the author to choose these specific IoU values?]

Response 3:[The reason for choosing 0.5 and 0.5:0.95 IoU thresholds in the AP calculation is as follows: AP50(IoU threshold of 0.5) and AP50-95 (coco AP) are used as common metrics to compare algorithm performance, and all have performance comparison sessions for this metric in related papers with RCNN/YOLO/DETR. Using these metrics makes it easier to compare the performance of different models.

In the scenario of this paper, since vehicle detection often involves small and tiny targets, the algorithm is more difficult to predict the frame of very small targets, so mAP50 is a more suitable indicator for comparison, and the purpose of comparing mAP50-95(coco AP), is to provide feedback on the prediction accuracy of the target frame.

In addition, algorithms other than RT-DETR use 60% of IoU as a threshold for nms, and this threshold is the default threshold for the YOLO algorithm to facilitate comparison with other algorithms.]